# Correlates of cardiorespiratory fitness in a population-based sample of middle-aged adults: cross-sectional analyses in the SCAPIS study

Mats Börjesson,[1,2] Örjan Ekblom,[3] Daniel Arvidsson,[4] Emerald G Heiland,[3,5] Daniel Väisänen,[3] Göran Bergström,[6,7] Elin Ekblom-Bak  [3]

For numbered affiliations see end of article.

**Correspondence to**
Dr Elin Ekblom-Bak;
eline@gih.se

## ABSTRACT

**Objectives** This study aimed to identify main sex-specific correlates of cardiorespiratory fitness (CRF) in a population-based, urban sample of Swedish adults.

**Design** Cross-sectional.

**Setting** Multi-site study at university hospitals, data from the Gothenburg site.

**Participants** A total of 5308 participants (51% women, aged 50–64 years) with a valid estimated $VO_2$max, from submaximal cycle test, in the Swedish CArdioPulmonary bioImage Study (SCAPIS), were included.

**Primary and secondary outcomes** A wide range of correlates were examined including (a) sociodemographic and lifestyle behaviours, (b) perceived health, anthropometrics and chronic conditions and (c) self-reported as well as accelerometer-derived physical activity and sedentary behaviours. Both continuous levels of estimated $VO_2$max as well as odds ratios (OR) and confidence intervals (CI)s of low $VO_2$max (lowest sex-specific tertile) were reported.

**Results** In multivariable regression analyses, higher age, being born abroad, short education, high waist circumference, poor perceived health, high accelerometer-derived time in sedentary and low in vigorous physical activity, as well as being passive commuter, correlated independently and significantly with low $VO_2$max in both men and women (OR range 1.31–9.58). Additionally in men, financial strain and being an ex-smoker are associated with higher odds for low $VO_2$max (OR 2.15; 95% CI 1.33 to 3.48 and OR 1.40; 95% CI 1.09 to 1.80), while constant stress with lower odds (OR 0.61; 95% CI 0.43 to 0.85). Additionally in women, being a regular smoker is associated with lower odds for low $VO_2$max (OR 0.64; 95% CI 0.45 to 0.92).

**Conclusions** The present study provides important reference material on CRF and correlates of CRF in a general middle-aged population, which can be valuable for future research, clinical practice and public health work. If relations are causal, increased knowledge about specific subgroups will aid in the development of appropriate, targeted interventions.

## BACKGROUND

Levels of cardiorespiratory fitness (CRF), mainly determined by recent levels of

## STRENGTHS AND LIMITATIONS OF THIS STUDY

⇒ The large population-based sample with estimated $VO_2$max from a submaximal test and accelerometer-measured physical activity provides the power required to analyse multiple relevant correlates of cardiorespiratory fitness.

⇒ Several high-risk subgroups with significantly lower estimated $VO_2$max were identified.

⇒ However, the cross-sectional design of the study prevents any causal inference, and we have no information on any genetic contribution to variation between subgroups of correlates.

⇒ Although based on a large sample from the general population, participants included were younger, leaner, more physically active and had more often a university degree compared with excluded participants.

moderate-to-vigorous physical activity and genetics,[1] is important for aerobic performance and engagement in daily exercise activities. It is also a strong predictor of several non-communicable diseases and all-cause mortality.[2–4] Unfortunately, data indicate a decline in CRF on population levels over the last decades.[5 6] CRF is typically lower in older ages,[7] in individuals with shorter education,[8] daily smokers and obese persons, yet higher in those engaging regularly in leisure time physical activity (PA) or exercise.[9] However, a broader knowledge of key determinants for CRF level is scarce, especially from population-based samples. For example, whether and how CRF levels vary by psychosocial factors and types of domain-specific PA behaviour are not fully elucidated or show conflicting results.[9 10] Moreover, previous studies of PA correlates on CRF are often limited by the use of self-reported PA.[11] Also, it is still not fully explored whether and how the age-related lower levels of CRF are associated with less exercise performed, or

instead is due to age-related changes in key physiological capacities itself.[12] Previous studies have mainly been based on selected samples, consisting of smaller cohorts (n<2000), which limits the power for comparisons between different subgroups.[9] More detailed knowledge of current correlates in unselected samples is needed to understand variations of CRF in the population, not only for the development of targeted assessments of CRF in the population but also the generation of appropriate interventions.

The aim of the present study is to identify main sex-specific correlates of CRF in a population-based, urban sample of 50–64 year-old men and women, including a wide range of correlates: (a) sociodemographic factors and lifestyle behaviours, (b) perceived health, anthropometrics and chronic conditions and (c) self-reported as well as accelerometer-derived PA and sedentary behaviours. A secondary aim is to study whether the commonly reported lower CRF with older age might be influenced by level of accelerometer-derived sedentary time and/or vigorous PA.

## METHODS

Data were retrieved from the Swedish CArdioPulmonary bioImage Study (SCAPIS).[13] In 2012, a comprehensive pilot study was conducted and in 2013–2018 the full SCAPIS study was carried out in collaboration with six Swedish university hospitals (Gothenburg, Linköping, Malmö/Lund, Stockholm, Umeå and Uppsala), aiming to recruit 5000 individuals (men and women, 50–64 years) from their respective municipality. Data collection took place over two to three occasions within a 2-week period. The participants underwent extensive imaging and functional studies of the heart, lungs and metabolism, filled in an extensive questionnaire, and wore an accelerometer during 7 days. At each test-site, there was a possibility to include site-specific measurements. At the Gothenburg site, a submaximal cycle test was performed to estimate VO$_2$max. In the pilot study, a total of 2243 participants were recruited from the census register, and 1111 (49.5%) agreed to participate. In the full study in Gothenburg, 12 109 were recruited and 6266 (52%) participated. Thus, a total of 7377 were eligible and 5308 participants (72%) provided valid estimated VO$_2$max.

### Assessment of cardiorespiratory fitness

CRF was assessed as estimated VO$_2$max using the Ekblom-Bak submaximal cycle ergometer test.[14] The test uses the difference in heart rate response between 4 min cycling on a lower, standardised work rate (30 watts) and 4 min cycling on a higher, individually chosen work rate (aiming at 13–14 at the Borg Rating of Perceived Exertion scale). Pedal frequency is held constant at both work rates (60 revolutions per minute) and heart rate is measured as the mean during the last minute at each work rate. The heart rate difference is then related to the increase in work rate between

the two work rates, and introduced in sex-specific validated algorithms for estimation of VO$_2$max. The test has shown high validity in reference to direct measurement of VO$_2$max in this age-group ($r^2$=0.84, SE of estimate 0.33 L/min).[14] Submaximal testing of CRF has been shown to predict all-cause mortality and cardiovascular disease (CVD) events in large, contemporary populations,[15 16] which highlights the possibility to use submaximal testing in larger population samples. To minimise well-known errors with submaximal testing, participants were requested to refrain from consuming a heavy meal or drinking coffee during the hours before the test. A priori exclusion criteria included symptoms of ongoing infections, known unstable CVD, electrocardiography patterns indicative of cardiac disease, use of beta-blockers, weight >125 kg or resting heart rate of >100 bpm (n=466). Moreover, participants did not take part in the testing if refrained from participating in testing (n=1209), did not receive a valid fitness test due to lack of test criteria fulfilment, did not fulfil the test or malfunction of heart rate monitors or the ergometer (n=394).

### Assessment of anthropometrics and chronic conditions

Measurements of weight, height and waist circumference were assessed during the first visit to the test centre. Body mass index (BMI) was calculated (kg/m$^2$), and waist circumference was measured at the midpoint between the top of the iliac crest and the lower margin of the lowest palpable rib. Prevalent depressive symptoms lasting 2 weeks or more during the last 12 months were self-reported. Prevalent chronic conditions were defined as reporting diagnosed disease or surgical treatment for CVD (ie, myocardial infarction, angina pectoris, stroke, congestive heart failure, atrial fibrillation), hypertension, lung disease, dyslipidaemia, diabetes, rheumatic disease and/or cancer.

### Assessment of physical activity pattern

Self-reported PA included commuting habits, physical working situation, exercise, leisure time sitting and total PA (see online supplemental additional file 1). Sedentary behaviour and PA were also derived from triaxial accelerometers (ActiGraph model GT3X and GT3X+, ActiGraph, Pensacola, Florida, USA). Participants were instructed to wear the accelerometer on an elastic belt over the right hip during all waking hours for at least seven consecutive days, except during water-based activities. ActiLife V.6.10.1 software was used to initialise the accelerometers and to download and process the collected data. The accelerometer recorded raw data (30 Hz) from all three axes, which were combined into a resulting vector, and extracted as 60 s epochs using a low frequency extension filter. Sedentary was defined as <200 counts per minute (cpm),[17] low intensity PA as 200–2689 cpm, moderate intensity PA as 2690–6166 cpm and vigorous PA (VPA) as ≥6167 cpm.[18] Non-wear time was defined as ≥60

**Table 1** Estimated VO$_2$max (mL/kg/min) in association with sociodemographic factors and lifestyle behaviour correlates

| | n | Estimated VO$_2$max Median (95% CI) | Low estimated VO$_2$max % within subgroup | OR* (95% CI) |
|---|---|---|---|---|
| **Sex** | | | | |
| Men | 2590 | 36.2 (35.8 to 36.5)† | na‡ | na |
| Women | 2718 | 30.4 (30.1 to 30.7) | na | na |
| **Age** | | | | |
| 50–54 years | 1881 | 34.9 (34.5 to 35.1)§ | 24 | 1 (ref) |
| 55–59 years | 1812 | 33.7 (33.2 to 34.0) | 32 | 1.46 (1.27 to 1.69) |
| 60–64 years | 1615 | 31.5 (31.0 to 31.8) | 45 | 2.57 (2.23 to 2.97) |
| **Marital status** | | | | |
| Married/cohabitation | 3766 | 33.8 (33.6 to 34.1)† | 31 | 1 (ref) |
| Divorced/single/widower | 1461 | 32.1 (31.7 to 32.6) | 38 | 1.37 (1.21 to 1.56) |
| **Born in Sweden** | | | | |
| Yes | 4341 | 33.7 (33.4 to 33.9)† | 31 | 1 (ref) |
| No | 923 | 32.0 (31.3 to 32.5) | 43 | 1.75 (1.51 to 2.03) |
| **Educational level** | | | | |
| University degree | 2480 | 34.3 (33.9 to 34.6)§ | 26 | 1 (ref) |
| High school/vocational education | 2215 | 32.8 (32.4 to 33.2) | 38 | 1.85 (1.63 to 2.10) |
| Elementary school | 555 | 31.6 (30.8 to 32.4) | 50 | 2.82 (2.32 to 3.42) |
| **Employment** | | | | |
| Working, ≥50% of full-time | 4342 | 33.9 (33.6 to 34.1)§ | 30 | 1 (ref) |
| Retired | 231 | 30.3 (29.8 to 31.4) | 53 | 1.55 (1.17 to 2.05) |
| Disability pension/sickness pension/sick leave | 299 | 29.3 (28.4 to 30.2) | 54 | 2.60 (2.05 to 3.31) |
| Unemployed/student | 365 | 31.3 (30.9 to 32.2) | 45 | 1.88 (1.51 to 2.35) |
| **Financial strain** | | | | |
| No | 4759 | 33.6 (33.4 to 33.9)† | 32 | 1 (ref) |
| Yes | 406 | 30.1 (29.6 to 30.7) | 51 | 2.51 (2.04 to 3.10) |
| **Smoking habits** | | | | |
| Never smoker | 2463 | 34.6 (34.2 to 34.9)§ | 29 | 1 (ref) |
| Former smokers | 2015 | 32.3 (31.9 to 32.6) | 38 | 1.38 (1.21 to 1.57) |
| Regular smoker/sometimes | 716 | 32.2 (31.8 to 32.9) | 39 | 1.61 (1.35 to 1.92) |
| **Pack-years** | | | | |
| Never smokers (0) | 2463 | 34.6 (34.2 to 34.9) | 29 | 1 (ref) |
| Former smokers, low (<15 pack-years) | 1220 | 33.0 (32.6 to 33.7) | 30 | 1.00 (0.86 to 1.17) |
| Former smokers, heavy (≥15 pack-years) | 795 | 31.1 (30.5 to 31.6) | 49 | 2.17 (1.83 to 2.56) |
| Current smokers, low (<15 pack-years) | 226 | 33.6 (32.4 to 34.8) | 30 | 1.11 (0.82 to 1.50) |
| Current smokers, heavy (≥15 pack-years) | 490 | 31.8 (31.1 to 32.3) | 42 | 1.88 (1.53 to 2.30) |
| **Alcohol use** | | | | |
| No/low | 3948 | 33.7 (33.4 to 34.0) | 30 | 1 (ref) |
| Moderate | 639 | 33.3 (33.0 to 33.9) | 34 | 1.26 (1.05 to 1.51) |
| High | 33 | 32.5 (30.0 to 33.9) | 52 | 3.06 (1.51 to 6.19) |

*Adjusted for sex and age, comparing the lowest sex-specific tertile of estimated VO$_2$max with the two higher tertiles.
†Significantly different from other subgroup (Mann-Whitney U test, p<0.0001).
‡Percentage in low estimated VO$_2$max and OR not applicable for comparison between men and women, as tertiles were sex-specific.
§Kruskal-Wallis analysis of variance p<0.0001.

consecutive minutes with no movement (0 cpm), with allowance for maximum 2 min of counts between 0 and 200 cpm. Wear time was calculated as 24 hours minus non-wear time. A minimum of 600 min of valid daily wear time for at least 4 days was required for inclusion.[19] Prolonged sedentary time was defined as ≥20 min of cpm below 200, with no allowance for interruption above threshold.

**Table 2** Estimated VO$_2$max (mL/kg/min) in association with perceived health, anthropometrics and chronic condition correlates

| | | Estimated VO$_2$max | | Low estimated VO$_2$max | |
|---|---|---|---|---|---|
| | n | Median (95% CI) | | % within subgroup | OR* (95% CI) |
| Sleep | | | | | |
| Very well/well | 2401 | 34.2 (33.9 to 34.5)† | | 31 | 1 (ref) |
| Rather well/badly/very badly | 2835 | 32.6 (32.2 to 32.9) | | 35 | 1.22 (1.09 to 1.38) |
| Stress | | | | | |
| No stress/some stress during last 5 years | 4089 | 33.7 (33.4 to 33.9)† | | 33 | 1 (ref) |
| Constant stress last 1–5 years | 1129 | 32.2 (31.8 to 32.6) | | 36 | 1.25 (1.08 to 1.44) |
| Control at work | | | | | |
| Strongly agree/agree | 3846 | 33.8 (33.6 to 34.1)† | | 31 | 1 (ref) |
| Neutral/do not agree/do not agree at all | 1111 | 32.3 (31.9 to 32.8) | | 35 | 1.23 (1.06 to 1.42) |
| Control in life | | | | | |
| Do not agree at all/do not agree | 2976 | 34.0 (33.7 to 34.3)† | | 30 | 1 (ref) |
| Neutral/agree/strongly agree | 2193 | 32.5 (32.1 to 32.8) | | 37 | 1.46 (1.29 to 1.64) |
| General health | | | | | |
| Excellent/very good | 2698 | 35.0 (34.8 to 35.3)‡ | | 23 | 1 (ref) |
| Good | 1818 | 32.5 (32.2 to 33.0) | | 40 | 2.41 (2.11 to 2.75) |
| Somewhat bad/bad | 729 | 29.4 (28.8 to 30.1) | | 56 | 4.80 (4.02 to 5.73) |
| Body mass index with estimated VO$_2$max in L/min | | | | | |
| <18 | 11 | 2.09 (1.79 to 2.55)‡ | | 55 | 0.65 (0.19 to 2.29) |
| 18.0–24.9 | 2025 | 2.38 (2.35 to 2.40) | | 36 | 1 (ref)§ |
| 25.0–29.9 | 2341 | 2.75 (2.70 to 2.80) | | 33 | 2.32 (1.95 to 2.76) |
| 30.0–34.9 | 728 | 2.63 (2.52 to 2.73) | | 29 | 5.79 (4.30 to 7.81) |
| ≥35 | 203 | 2.38 (2.29 to 2.49) | | 29 | 16.6 (10.2 to 27.1) |
| Waist circumference | | | | | |
| Low waist (<88 cm women, <102 cm men) | 3191 | 36.4 (36.2 to 36.7)† | | 15 | 1 (ref) |
| High waist (≥88 cm women, ≥102 cm men) | 2114 | 28.9 (28.6 to 29.2) | | 61 | 10.2 (8.8 to 11.7) |
| Depression symptoms | | | | | |
| No | 3687 | 33.9 (33.7 to 34.2)† | | 32 | 1 (ref) |
| Yes | 1495 | 32.0 (31.6 to 32.3) | | 37 | 1.33 (1.17 to 1.51) |
| Chronic conditions¶ | | | | | |
| 0 | 3465 | 33.9 (33.7 to 34.3)‡ | | 30 | 1 (ref) |
| 1–2 | 1705 | 32.3 (31.9 to 32.8) | | 39 | 1.38 (1.22 to 1.57) |
| ≥3 | 138 | 31.0 (29.7 to 32.7) | | 51 | 2.09 (1.48 to 2.96) |
| Cardiovascular disease, hypertension and/or lung disease | | | | | |
| No | 4069 | 33.7 (33.5 to 34.0) | | 32 | 1 (ref) |
| Yes | 1239 | 32.0 (31.5 to 32.5) | | 37 | 1.42 (1.24 to 1.62) |

*Adjusted for sex and age, comparing the lowest sex-specific tertile of estimated VO$_2$max with the two higher tertiles.
†Significantly different from other subgroup (Mann-Whitney U test, p<0.0001).
‡Kruskal-Wallis analysis of variance p<0.0001.
§Sex-specific tertiles based on L/min. Analyses additionally adjusted for weight in kg.
¶Includes cardiovascular disease, hypertension, lung disease, dyslipidaemia, diabetes, rheumatic disease and cancer.

## Assessment of sociodemographic, lifestyle and perceived symptoms and health

Marital status, country of birth, educational level, employment status, financial strain and smoking status were self-reported and grouped according to definitions presented in table 1. Pack-years of smoking was derived by multiplying the average number of cigarettes smoked per day by the number of years smoking, divided by the number of cigarettes per package. The 10-item screening tool Alcohol Use Disorders Identification Test (AUDIT) was used to define alcohol abuse,[20] and categorised into three groups, 'No/Low', 'Moderate' and 'High', according to the AUDIT score 0–7, 8–15 and >15 for men, and 0–5, 6–13 and >13 for

**Table 3** Estimated VO$_2$max (mL/kg/min) in association with self-reported and accelerometer-derived physical activity pattern correlates

| | n | Estimated VO$_2$max Median (95% CI) | Low estimated VO$_2$max % within subgroup | OR$^C$ (95% CI) |
|---|---|---|---|---|
| Self-report | | | | |
| Commuting habits* | | | | |
| No active commuting | 3057 | 33.1 (32.8 to 33.4)† | 37 | 1 (ref) |
| Any active commuting, further divided into: | 1738 | 34.6 (34.2 to 35.0) | 23 | 0.47 (0.41 to 0.54) |
| Partly active commuter, bike | 671 | 34.8 (34.1 to 35.5) | 20 | 0.41 (0.33 to 0.50) |
| Partly active commuter, walking | 196 | 30.4 (29.3 to 31.6) | 41 | 1.12 (0.83 to 1.52) |
| Partly active commuter, mix | 19 | 30.2 (28.8 to 33.5) | 21 | 0.45 (0.15 to 1.38) |
| All year active commuter, bike | 407 | 38.1 (36.9 to 39.2) | 10 | 0.18 (0.13 to 0.25) |
| All year active commuter, walking | 415 | 33.0 (32.2 to 33.4) | 31 | 0.71 (0.57 to 0.89) |
| All year active commuter, mix | 30 | 34.7 (30.6 to 37.3) | 23 | 0.51 (0.22 to 1.22) |
| Physical working situation | | | | |
| Sedentary to light | 3921 | 33.7 (33.5 to 34.1)‡ | 30 | 1 (ref) |
| Sometimes to frequently heavy | 945 | 32.5 (32.0 to 33.1) | 38 | 1.44 (1.24 to 1.67) |
| Exercise habits | | | | |
| Never | 1384 | 31.0 (30.7 to 31.4)† | 51 | 1 (ref) |
| Irregular | 1067 | 31.3 (30.8 to 31.7) | 43 | 0.75 (0.64 to 0.88) |
| 1–2 times per week | 1032 | 33.5 (33.0 to 34.1) | 27 | 0.35 (0.29 to 0.41) |
| 2–3 times per week | 1089 | 35.2 (34.8 to 35.7) | 20 | 0.25 (0.21 to 0.30) |
| >3 times per week | 649 | 38.3 (37.8 to 38.9) | 12 | 0.14 (0.11 to 0.19) |
| Total physical activity | | | | |
| Sedentary | 540 | 30.0 (29.1 to 30.5)† | 63 | 1 (ref) |
| Light | 2366 | 31.5 (31.2 to 31.9) | 42 | 0.39 (0.32 to 0.48) |
| Moderate | 1613 | 34.9 (34.6 to 35.4) | 21 | 0.14 (0.11 to 0.18) |
| Regular exercise/training | 669 | 38.6 (38.0 to 39.1) | 9 | 0.06 (0.05 to 0.09) |
| Leisure time sitting | | | | |
| Q1; <4 hours per day | 1121 | 34.1 (33.5 to 34.7) | 28 | 1 (ref) |
| Q2; 4–6 hours per day | 1039 | 33.6 (33.0 to 34.2) | 32 | 1.24 (1.02 to 1.49) |
| Q3; 6–8.5 hours per day | 925 | 34.0 (33.5 to 34.5) | 29 | 1.11 (0.91 to 1.36) |
| Q4; >8.5 hours per day | 958 | 33.7 (33.3 to 34.2) | 32 | 1.31 (1.08 to 1.59) |
| % of time spent in sedentary | | | | |
| Q1; <47% | 1267 | 34.1 (33.5 to 34.5)† | 26 | 1 (ref) |
| Q2; 47%–54% | 1265 | 33.6 (33.1 to 34.1) | 28 | 1.19 (0.99 to 1.42) |
| Q3; 54%–61% | 1266 | 33.4 (33.0 to 33.7) | 33 | 1.52 (1.27 to 1.81) |
| Q4; >61% | 1267 | 33.0 (32.3 to 33.3) | 43 | 2.44 (2.05 to 2.91) |
| % of time spent in prolonged sedentary (of total wear time) | | | | |
| Q1; <13% | 1266 | 34.1 (33.6 to 34.5)† | 25 | 1 (ref) |
| Q2; 13%–19% | 1267 | 33.5 (33.0 to 34.0) | 28 | 1.14 (0.96 to 1.37) |
| Q3; 19%–26% | 1265 | 33.5 (33.1 to 34.0) | 35 | 1.60 (1.35 to 1.91) |
| Q4; >26% | 1267 | 32.7 (32.2 to 33.1) | 42 | 2.18 (1.83 to 2.60) |
| % of time spent in light physical activity | | | | |
| Q1; <33% | 1267 | 33.7 (33.2 to 34.0) | 39 | 1 (ref) |
| Q2; 33%–40% | 1266 | 33.3 (32.9 to 33.7) | 34 | 0.78 (0.66 to 0.92) |
| Q3; 40%–46% | 1265 | 33.5 (33.1 to 34.0) | 29 | 0.60 (0.51 to 0.72) |
| Q4; >46% | 1267 | 33.2 (32.8 to 33.8) | 28 | 0.56 (0.47 to 0.66) |
| % of time spent in moderate physical activity | | | | |

**Table 3** Continued

| | n | Estimated VO$_2$max | Low estimated VO$_2$max | |
|---|---|---|---|---|
| | | Median (95% CI) | % within subgroup | OR$^C$ (95% CI) |
| Q1; <3.8% | 1267 | 31.6 (31.6 to 32.5)† | 44 | 1 (ref) |
| Q2; 3.8%–5.3% | 1265 | 33.3 (32.7 to 33.7) | 32 | 0.62 (0.52 to 0.73) |
| Q3; 5.3%–7.2% | 1266 | 33.8 (33.3 to 34.1) | 29 | 0.52 (0.44 to 0.62) |
| Q4; >7.2% | 1267 | 34.9 (34.3 to 35.3) | 26 | 0.44 (0.37 to 0.52) |
| % of time spent in vigorous physical activity | | | | |
| Q1; <0.02% | 1266 | 30.5 (30.1 to 31.0)† | 49 | 1 (ref) |
| Q2; 0.02%–0.1% | 1266 | 31.7 (31.3 to 32.1) | 43 | 0.80 (0.69 to 0.94) |
| Q3; 0.1%–0.6% | 1267 | 34.3 (34.0 to 34.7) | 23 | 0.34 (0.28 to 0.40) |
| Q4; >0.6% | 1266 | 36.7 (36.3 to 37.1)† | 16 | 0.21 (0.18 to 0.26) |

$^C$Adjusted for sex and age, comparing the lowest sex-specific tertile of estimated VO2maxwith the two higher tertiles.
*Analysed both as comparing No active commuting and Any active commuting, as well as further dividing Any active commuting into; Partly commuting, defined as going by bike or walking at least one season out of four; All year commuting, defined as going by bike or walking all year around. Partly and all year commuting 'walking' or 'bike' were defined as the majority of seasons commuted with that mode of active commuting. Mix was defined when the seasons of active commuting were spent equally between the two modes.
†Kruskal-Wallis analysis of variance p<0.0001.
‡Significantly different from other subgroup (Mann-Whitney U test, p<0.0001).

women. Sleep, stress, perceived control at work and in leisure-time, as well as general health, were also self-reported and grouped as presented in table 2.

## Patient and public involvement
No patient involved.

## Statistical analysis
CRF was skewed (Kolmogorov-Smirnov test), hence, descriptive data are presented as medians and 95% CIs. Low CRF was defined as the lowest sex-specific tertile (cut-off 27.75 mL/kg/min for women and 33.65 mL/kg/min for men). ORs with 95% CIs for low CRF in relation to the subgroups of the correlates were calculated using binary logistic regression modelling adjusting for sex and age. Interaction between sex and each correlate was assessed by adding an interaction term to the model, and significant interactions were defined as p<0.05 for the interaction term. To study independent associations of correlates with low CRF, all correlates not displaying multicollinearity (Spearman's ρ>0.6) were entered in a backward logistic multivariable regression model, with low CRF as the dependent variable. To study whether the relationship between age-group and estimated VO$_2$max was modified by VPA or prolonged sedentary, general linear modelling with post hoc analyses was used including an interaction term for age-group×VPA or age-group×prolonged sedentary, and adjusting for sex. The VPA variable was divided into four groups: 0 min/week, 0 to 37.5 min/week (half of the recommended VPA level), >37.5 to <75 min/week (the recommended VPA level) and ≥75 min/week.[21] For sedentary behaviour, the quartiles from table 3 were used. Statistical analyses were performed using SPSS (V.26.0, 2019, SPSS, Chicago, Illinois, USA) and R (R V.4.1.1) with R packages ggplot2 and ggeffects.

## RESULTS
Of the total 7377 participants eligible for exercise testing in the study, 5308 provided valid estimates for VO$_2$max. Participants included, compared with participants not included (n=2069), were more often women (54% vs 51%, p=0.040), younger (age 57.2 vs 58.2 years, p<0.001), had lower BMI (26.5 vs 27.7 kg/m$^2$, p<0.001), had more often university degree (47% vs 35%, p<0.001) and more often reported to be weekly regular exercisers (53% vs 40%, p<0.001). Additionally, among included participants, 14% were regular smokers, 2.3% had established CVD, 17% had hypertension, 11% had dyslipidaemia and 2.5% had diabetes (all self-reported).

Associations between CRF and the different correlates are presented in tables 1–3 and online supplemental appendix figures 1 and 2, with sex-specific analyses presented in online supplemental appendix table 1. The results can be summarised as follows:

Sociodemographic factors and lifestyle behaviours (table 1): Women, older age, divorced/single/widowed, participants not born in Sweden, lower education, employment <50% of full-time/retired/pension/unemployed, with financial strain or being regular and former smokers had lower estimated CRF and higher OR for having low CRF, compared with their counterparts. Older men (60–64 years) were more likely to have lower CRF than older women (p=0.045), with no other interactions seen between sex and the other sociodemographic variables. For lifestyle factors, there were significant interactions between sex and smoking, such that men being regular smokers/sometimes smokers, and with more pack-years

in life, were more likely to have low CRF (p=0.004 and p=0.008) compared with smoking women.

Perceived health, anthropometrics and chronic conditions (table 2 and online supplemental appendix figure 1): Participants reporting poor sleep, high stress, low control at work, low perceived degree of control in life, poorer general health, having higher BMI, higher waist circumference, prevalence of depression symptoms and higher number of chronic conditions had lower estimated CRF. Women with low perceived degree of control in life were more likely to have low CRF than their male

counterparts (p=0.004). Higher BMI and higher waist circumference were more strongly associated with low CRF in women than in men (p=0.001 and p=0.041).

Self-reported and accelerometer-derived PA pattern (table 3 and online supplemental appendix figure 2): Participants engaging in any active commuting had higher CRF compared with those with no active commuting. This was mainly seen among partly-year and all-year commuters by bike. Participants with a more strenuous physical working situation, less weekly exercise, lower self-reported total PA level and high

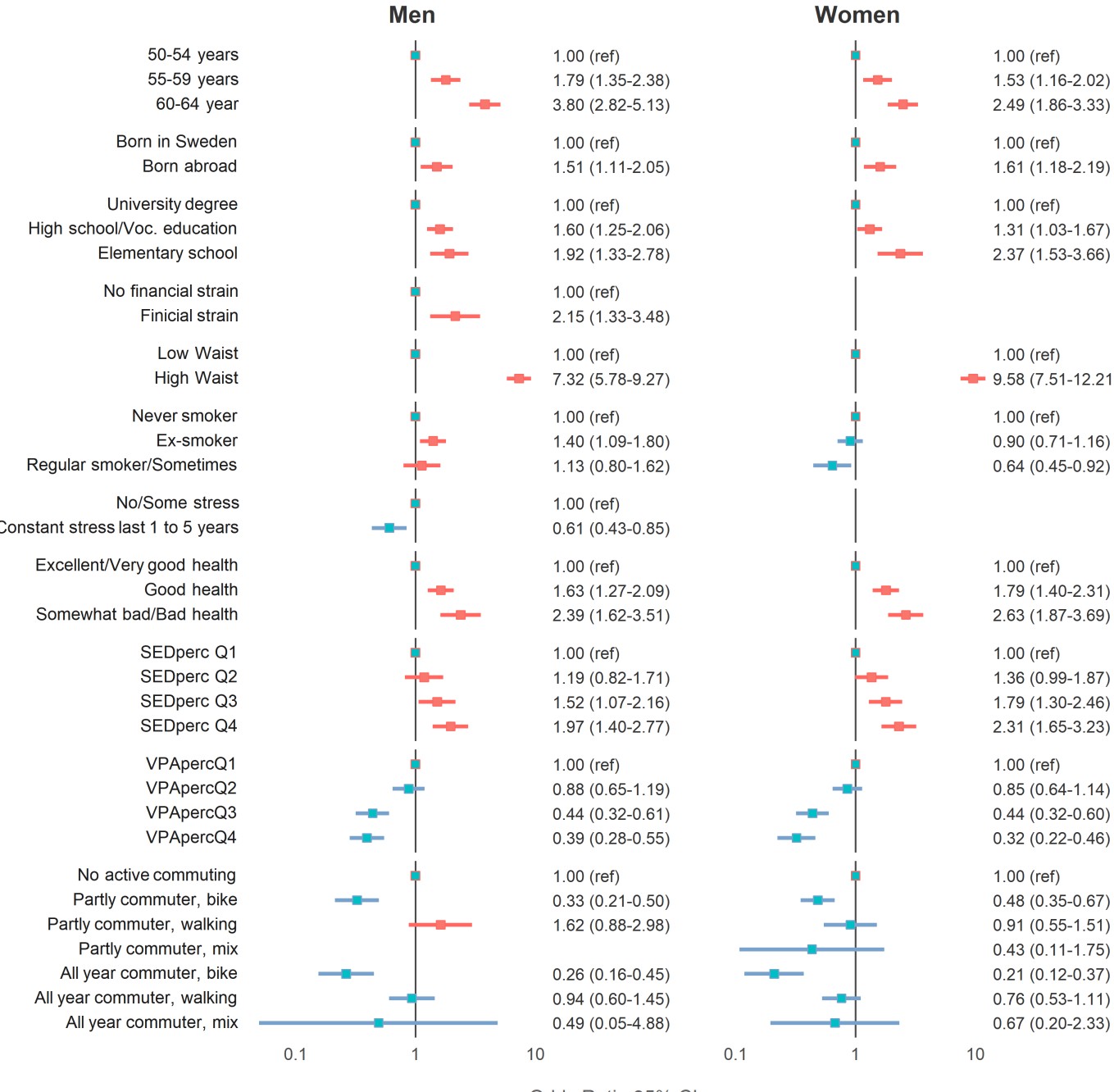

**Figure 1** ORs (95% CIs) for low cardiorespiratory fitness in association with correlates in the backward multiregression model. ORs are presented on log-scale. perc, per cent; Q, quartile; SED, sedentary; voc, vocational; VPA, vigorous physical activity.

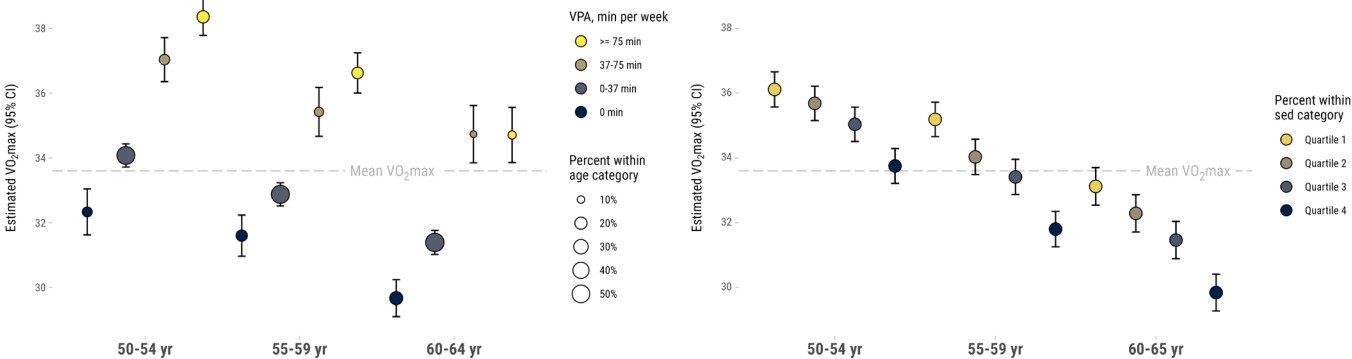

**Figure 2** Mean estimated VO$_2$max (95% CI) in association with age-group (x-axis) and vigorous physical activity (VPA) level in min/week (left) and percentage of time in sedentary (right).

self-reported sitting time had lower CRF. Women reporting higher total PA had lower OR for low CRF (p=0.021) compared with men. Higher accelerometer-derived time spent in sedentary and prolonged sedentary time were associated with lower CRF, whereas more time spent in low intensity PA, moderate PA and VPA were associated with higher CRF. Women with higher sedentary time were more likely to have lower CRF compared with men (p=0.033).

In the multivariable regression model (figure 1), older age-group, being born abroad, having lower education, high waist circumference, poorer general health and more time spent sedentary, were associated with a higher OR for low CRF in both men and women. More time spent in VPA and commuting by bike (partly and all-year) were associated with a lower OR. Specifically in men, financial strain and being an ex-smoker were associated with higher OR, and constant stress with a lower OR, for a low CRF. Specifically in women, being a regular smoker was associated with lower OR for low CRF, with no significant associations with financial strain or stress.

In figure 2, higher levels of VPA were more common in younger age-groups (left part of figure). However, higher levels of VPA are associated with significantly higher CRF in all age-groups (p<0.001). There was no overall interaction between VPA and age-group (p=0.372). Stratified analyses showed that there was no significant difference in CRF between participants with 37.5–75 min and ≥75 min of VPA per week in the oldest age-group (p=1.00). Also, participants aged 50–54 years and 55–59 years with 0 min/ week of VPA had similar CRF (p=0.305 for differences between groups), and so did participants aged 55–59 and 60–64 years with 37.5–75 min/week (p=1.000 for differences between groups). In the right part of figure 2, higher levels of time spent sedentary were associated with lower CRF in all age-groups. There was no overall interaction effect between sedentary time and age-group (p=0.596). Stratified analyses showed that there was some overlapping between age-groups and time sedentary for CRF.

## DISCUSSION

The present study identified multiple correlates of CRF in a large, population-based sample of middle-aged, urban Swedish adults from the SCAPIS study. The results confirm some previously documented correlates, but also adds important novel aspects, including correlates of depression symptoms, commuting habits and accelerometer-derived PA. Some of the most interesting correlates are discussed, as follows:

First, the strong, positive association between CRF and length of education confirms previous results,[8–10] while similar associations with being born in Sweden, being employed and having an economic buffer, adds novel information regarding sociodemographic aspects. These associations may largely be mediated by level of VPA. Individuals in these subgroups have previously been shown to engage in higher levels of daily VPA compared with their counterparts,[22] and a strong agreement between socioeconomic factors and both PA and CRF has been reported.[8] The association between lower CRF in groups with lower socioeconomic status is also in line with higher observed incidence of cardiovascular, mood-related disorders and premature mortality in lower socioeconomic strata compared with those of higher socioeconomic status.[23] Also, in a recent paper on risk of severe COVID-19, the associations between socioeconomic factors such as educational level, income and occupational group and severe COVID-19 risk, were largely mediated by the CRF level of the participants.[24]

Second, participants reporting more physically demanding work had in general a lower CRF, compared with more sedentary workers. While this is in line with previous reports from the Swedish working population,[25] this may at first be counterintuitive as higher occupational PA could be considered as a 'work-out' contributing to health benefits and higher CRF. However, although more physically demanding occupations induce a higher mean intensity compared with more sedentary occupations, the relative intensity of 'active jobs' have been shown to be maybe too low to

improve VO$_2$max.[26] Rather, the higher mean intensity may be tiring enough for the individual, which could lead to lower levels of fitness-enhancing leisure time VPA.[22] This, together with a more pronounced increase in body weight in blue-collar (more active) occupational groups,[27] may contribute to the general lower CRF levels in these groups. Also, the mean intensity during work has been associated with lower heart rate variability and a higher heart rate the following night.[26] This suggests an induced imbalance in the autonomic cardiac modulation, which may be one potential underlying mechanism for the detrimental effects on health proposed for occupational PA, also called the 'physical activity paradox'.[28] Thus, differences in CRF may partly contribute to the observed paradox.

Third, we could confirm the established relation between self-reported total and leisure-time PA and CRF,[9] however, expanding on knowledge regarding CRF in active commuters. Any active commuting was associated with higher CRF, however, the higher CRF was mainly driven by partly and all-year commuters by bike. This is line with higher levels of VPA shown in active commuters.[22] Although the present study cannot clarify whether it was the active commuting per se that contributed to higher CRF levels, mean intensity during normal commuting in habitual cycle commuters has been shown to be mainly constituted of the lower end of vigorous intensity[29] for longer activity bouts (>10 min). This then often adds up to recommended levels of 150 min per week.[30] Thus, regular active commuting by bike requires an at least moderate or even high CRF.

Fourth, previous data on correlates between CRF and accelerometer-assessed PA patterns are scarce and inconclusive.[9] We found that higher accelerometer-derived estimates of the different components of the PA pattern provided a strong association between CRF level and time in sedentary (negative association), and MPA and VPA (positive associations). Interestingly, both time in sedentary and in VPA remained significantly associated with CRF in the multivariable regression model, which implies separate importance of these intensities for CRF level.

Finally, previous research is unclear about whether higher PA level would ameliorate lower CRF levels among older individuals.[9] The cross-sectional analyses in our study showed that older individuals had lower CRF than younger, with strong associations between higher VPA and less time spent sedentary, and CRF in all age-groups. More time in VPA may be required for older individuals in order to maintain a certain CRF, as the mean estimated VO$_2$max in individuals 60–64 years old engaging in 37 min or more of VPA per week was overlapping with the VO$_2$max in individuals 50–54 years old doing less than 37 min of VPA per week. It is also possible that relations are reversed or at least bidirectional, that is, that a higher CRF will enable an individual to be physically active, including higher intensity activities. Indeed, previous studies have shown that those being active at higher age, have superior CRF compared with less active peers.[12] Age-related medico-physiological changes, possibly affecting the possibility to be active, such as loss of muscle mass or increasing concomitant disease and medications, may affect the intensity of the VPA performed.[7]

## Strengths and limitations

The cross-sectional design of the study prevents any causal inference and we cannot rule out reverse causality. We have no information on any genetic contribution to variation between subgroups of correlates. Although we have a large sample from the general population, those who participated in the fitness testing did differ from the source population. Therefore, translation of the results to populations that are less healthy and active and of other socioeconomic status (eg, lower education), should be done with caution. Further, the large sample size with estimated CRF from a submaximal test and accelerometer-measured PA provided the power required to analyse multiple correlates relevant for CRF. It is unknown if regular bicycling affects validity of a cycle ergometer test. Local adaptations to the thigh and gluteal muscles from regular cycling may alter systemic circulatory response to standard cycling exercise. If so, the finding that regular bike commuters had higher CRF may in part be ascribed to varying validity.

## CONCLUSIONS

This study in a large sample of middle-aged men and women delineates multiple modifiable as well as non-modifiable correlates of CRF. In addition, we identified specific subgroups as potential high-risk groups for CRF-related disorders, which constitute targets for behavioural change to increase CRF, if relations are causal. These groups include individuals with higher age, being born abroad, lower education, high waist circumference, poor perceived health, high accelerometer-derived time in sedentary and low in VPA, as well as passive commuters. Across all age-groups, higher VPA and less time spent sedentary, were associated with a higher CRF. Although having limited possibilities for causal inference, the present study provides important reference material of CRF and correlates in a general middle-aged population, which can be valuable for future research, clinical practice and public health work. Specifically, increased knowledge about specific subgroups will aid in the personalised prescription of exercise in healthcare. In light of the higher prevalence of low CRF in later years,[5] a great challenge remains in implementing effective interventions to support health in identified parts of the population.

**Author affiliations**
¹Center for Health and Performance, Department of Molecular and Clinical Medicine, University of Gothenburg, Gothenburg, Sweden

[2]Department of Medicine, Geraiatric and Acute Medicine Östra, Sahlgrenska University Hospital, Goteborg, Sweden

[3]Department of Physical Activity and Health, Swedish School of Sport and Health Sciences GIH, Stockholm, Sweden

[4]Center for Health and Performance, Department of Food and Nutrition, and Sport Science, University of Gothenburg, Gothenburg, Sweden

[5]Department of Surgical Sciences, Medical Epidemiology, Uppsala University, Uppsala, Sweden

[6]Sahlgrenska Center for Cardiovascular and Metabolic Research, Wallenberg Laboratory, Sahlgrenska University Hospital, Gothenburg, Sweden

[7]Department of Molecular and Clinical Medicine, University of Gothenburg, Gothenburg, Sweden

**Contributors** All authors contributed to the conception or design of the work. MB, ÖE, DA, DV and EE-B contributed to the acquisition, analysis or interpretation of data for the work. MB, ÖE, EGH, GB and EE-B drafted the manuscript or substantively revised it. All authors revised it critically for important intellectual content. All authors critically revised the manuscript, gave final approval and agreed to be accountable for all aspects of the work in ensuring that questions related to the accuracy or integrity of any part of the work are appropriately investigated and resolved. The corresponding author, EE-B, is the manuscripts guarantor and attests that all listed authors meet authorship criteria, that no others meeting the criteria have been omitted and take the full responsibility for the overall content.

**Funding** The main funding body of The Swedish CArdioPulmonary bioImage Study (SCAPIS) is the Swedish Heart and Lung Foundation (N/A). The study is also funded by the Knut and Alice Wallenberg Foundation (N/A), the Swedish Research Council (N/A) and VINNOVA (N/A) the University of Gothenburg and Sahlgrenska University Hospital (N/A), Karolinska Institutet and Stockholm county council (N/A), Linköping University and University Hospital (N/A), Lund University and Skåne University Hospital (N/A), Umeå University and University Hospital (N/A), Uppsala University and University Hospital (N/A). Author MB is supported by ALF grants from Västra Götalandsregionen (ALFGBG-965249) and a grant from the Heart and Lung Foundation (no 20210270). Author ÖE, EGH and EE-B were funded by grants from Skandia Risk & Hälsa (N/A). Author GB was supported by Heart and Lung Foundation (20180324), the Swedish Research Council (2019-01140) and LUA/ALF (ALFGBG-718851).

**Competing interests** None declared.

**Patient and public involvement** Patients and/or the public were not involved in the design, or conduct, or reporting, or dissemination plans of this research.

**Patient consent for publication** Consent obtained directly from patient(s).

**Ethics approval** The study was approved by the ethics board (Dnr 2010-228-31M, Dnr 2011-449-32M and Dnr 638-16) and adheres to the Declaration of Helsinki. All participants provided written informed consent.

**Provenance and peer review** Not commissioned; externally peer reviewed.

**Data availability statement** Data are available upon reasonable request. Due to the nature of the sensitive personal data and study materials, they cannot be made freely available. However, by contacting the corresponding author or study organisation (www.scapis.org), procedures for sharing data, analytical methods and study materials for reproducing the results or replicating the procedure can be arranged following Swedish legislation.

**ORCID iD**
Elin Ekblom-Bak http://orcid.org/0000-0002-3901-7833

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
