## [Reviewer comments · BMJ Open]

ARTICLE DETAILS

TITLE (PROVISIONAL)	Correlates of cardiorespiratory fitness in a population-based sample of middle-aged adults – cross-sectional analyses in the SCAPIS study
AUTHORS	Börjesson, Mats; Ekblom, Örjan; Arvidsson, D; Heiland, Emerald G; Väisänen, Daniel; Bergström, Göran; Ekblom-Bak, Elin

VERSION 1 – REVIEW

REVIEWER	Laukkanen, Jari A. Univ Eastern Finland, Institute of Clinical Medicine
REVIEW RETURNED	22-Jul-2022

GENERAL COMMENTS	This study provides very useful reference material for a general middle-aged population for clinical practice, and public health work. Increased knowledge about specific various populations will aid in the development of targeted interventions. The authors found that higher accelerometer-derived estimates of the different components of the physical activity (PA) pattern provided a strong association between CRF level and time in sedentary (negative association), and MPA and VPA (positive associations). 1. Please add data on common cardiovascular risk factors such as lipids, blood pressure, diabetes, and smoking.2. A submaximal cycle test was performed to estimate VO₂max. How was it actually estimated (based on heart rate or something else)?3. Were there subjects who were using medication such as beta-blockers, which may have an effect on heart rate responses?4. Cardiorespiratory fitness (CRF) is a strong and independent predictor for all-cause mortality as you have indicated in the introduction. However, please add the most updated and recent meta-analysis showing the association between CRF and mortality (PMID: 35562197); as it included more representative studies than older Kodama's publication in 2009. You may emphasize why should we assess CRF. Secondly, there are also studies showing that the CRF from the submaximal exercise test can be used to predict all-cause mortality in a large contemporary population (PMID: 32370851)5. Ekblom-Bak submaximal cycle ergometer test should be described shortly in the manuscript (irrespective of the already added reference)6. Have you assessed ECG and blood pressure using the exercise test? Who was supervising the exercise tests (safety reasons?)7. How many participants had any cardiovascular disease (myocardial infarction, angina pectoris, stroke, heart failure, atrial fibrillation)? What about stable coronary heart disease, was it diagnosed for some of the participants (or not)?
--

REVIEWER	Pentikäinen, Heikki Kuopio Research Institute of Exercise Medicine
REVIEW RETURNED	11-Oct-2022

GENERAL COMMENTS	General comments: I read with interest the manuscript “Correlates of cardiorespiratory fitness in middle-age – cross-sectional analyses in 5308 individuals in the SCAPIS study”. The paper is technically sound, and it is clearly written. The claims presented are supported by the data and they are appropriately discussed in the context of the previous literature. Important weakness is the cross-sectional design of the study, which is properly discussed by the authors. The full SCAPIS study was carried out in six Swedish university hospitals, but all submaximal cycle tests were performed at the Gothenburg site. I consider this as a strength of the study because it makes it less likely that the data would be somehow clustered. However, I have some worries and comments concerning mainly the results reported in Table 2. Participants reporting higher BMI had lower estimated VO₂max expressed as ml/kg/min and L/min. There is mathematical coupling problem present when VO₂max is expressed as ml/kg/min because body weight (kg) is included in both BMI and CRF. This doesn't make sense and should be removed from the results. Expressing VO₂max as L/min is better but considering that analysis is adjusted for weight (kg), it basically describes the association between height and CRF. Please consider whether it is rational to report this. The same mathematical coupling problem is present in the “Current weight compared to weight at 20 years of age” analysis. When VO₂max is expressed as ml/kg/min, the body weight (kg) is included in both the predictor and outcome variable. Although being a typical challenge in epidemiological studies, I also doubt how well these middle-aged participants can remember their body weight at 20 years of age. If authors are willing to report this, I recommend to express VO₂max as L/min. All these findings include the same common feature: the higher is the body weight the higher is the odds ratio for having low CRF (ml/kg/min). The same exact problem is not present when studying the association between waist circumference (cm) and the risk of low CRF. However, this emphasizes the same fact that obese people tend to have low CRF. Other comments: Constant stress in men, and being a regular smoker in women, associated with lower risk of low VO₂max. Could you please provide some discussion in relation to these findings? Are they just coincidence or could there be any other explanations for these contradictory findings? Please provide explanations for the symbols a,b and c in the footnotes of the Table 3.
--

VERSION 1 – AUTHOR RESPONSE

Reviewer: 1

Dr. Jari A. Laukkanen, Univ Eastern Finland

Authors: Thank you for your time and effort to help us improve the manuscript. We have replied to your comments point by point below. Changes in the manuscript are highlighted with track changes.

Comments to the Author:

This study provides very useful reference material for a general middle-aged population for clinical practice, and public health work. Increased knowledge about specific various populations will aid in the development of targeted interventions. The authors found that higher accelerometer-derived estimates of the different components of the physical activity (PA) pattern provided a strong association between CRF level and time in sedentary (negative association), and MPA and VPA (positive associations).

1. Please add data on common cardiovascular risk factors such as lipids, blood pressure, diabetes, and smoking.

Authors: Thank you for a relevant question. We have now added some additional characteristics of the study population in the beginning of the result section (page 9, line 210-212).

2. A submaximal cycle test was performed to estimate VO₂max. How was it actually estimated (based on heart rate or something else)?

Authors: We have now added information regarding the test in the manuscript (page 5-6, line 122-128).

3. Were there subjects who were using medication such as beta-blockers, which may have an effect on heart rate responses?

Authors: Thank you for a relevant question. Yes, as stated in the method section (page 6); "A priori exclusion criteria included symptoms of on-going infections, known unstable cardiovascular disease, electrocardiography patterns indicative of cardiac disease, use of beta-blockers, weight > 125 kg or resting heart rate of > 100 bpm (n=466)."

4. Cardiorespiratory fitness (CRF) is a strong and independent predictor for all-cause mortality as you have indicated in the introduction. However, please add the most updated and recent meta-analysis showing the association between CRF and mortality (PMID: 35562197); as it included more representative studies than older Kodama's publication in 2009. You may emphasize why should we assess CRF. Secondly, there are also studies showing that the CRF from the submaximal exercise test can be used to predict all-cause mortality in a large contemporary population (PMID: 32370851)

Authors: Thank you for the recommended publications. We have now added the first one to the introduction section (reference no 4), and the second one in the method section (page 6, line 130-132).

5. Ekblom-Bak submaximal cycle ergometer test should be described shortly in the manuscript (irrespective of the already added reference)

Authors: We have now described the test procedure in more detail in the method section (page 5-6, line 123-128).

6. Have you assessed ECG and blood pressure using the exercise test? Who was supervising the exercise tests (safety reasons?)

Authors: Thank you for a relevant question. The test was performed by trained nurses. However, the Ekblom Bak test is a submaximal test commonly used in health screenings and studies outside the health care system. It is considered a low-risk test, as the highest exercise intensity aims at a perceived exertion no higher than 14 on the Borg scale ("somewhat hard"), which corresponds to

moderate intensity. This level is considered low risk in asymptomatic patients even with CAD (according to ESC sports cardiology guidelines 2020).

7. How many participants had any cardiovascular disease (myocardial infarction, angina pectoris, stroke, heart failure, atrial fibrillation)? What about stable coronary heart disease, was it diagnosed for some of the participants (or not)?

Authors: As now stated in the beginning of the result section, approximately 2% of the included participants report previous diagnose of CVD. Participants with unstable CVD were excluded a priori from fitness testing.

Reviewer: 2

Dr. Heikki Pentikäinen, Kuopio Research Institute of Exercise Medicine, University of Eastern Finland Faculty of Health Sciences

Authors: Thank you for your time and effort to help us improve the manuscript. We have replied to your comments point by point below. Changes in the manuscript are highlighted with track changes.

Comments to the Author:

General comments:

I read with interest the manuscript "Correlates of cardiorespiratory fitness in middle-age – cross-sectional analyses in 5308 individuals in the SCAPIS study". The paper is technically sound, and it is clearly written. The claims presented are supported by the data and they are appropriately discussed in the context of the previous literature. Important weakness is the cross-sectional design of the study, which is properly discussed by the authors. The full SCAPIS study was carried out in six Swedish university hospitals, but all submaximal cycle tests were performed at the Gothenburg site. I consider this as a strength of the study because it makes it less likely that the data would be somehow clustered.

However, I have some worries and comments concerning mainly the results reported in Table 2.

Participants reporting higher BMI had lower estimated VO₂max expressed as ml/kg/min and L/min. There is mathematical coupling problem present when VO₂max is expressed as ml/kg/min because body weight (kg) is included in both BMI and CRF. This doesn't make sense and should be removed from the results. Expressing VO₂max as L/min is better but considering that analysis is adjusted for weight (kg), it basically describes the association between height and CRF. Please consider whether it is rational to report this.

The same mathematical coupling problem is present in the "Current weight compared to weight at 20 years of age" analysis. When VO₂max is expressed as ml/kg/min, the body weight (kg) is included in both the predictor and outcome variable. Although being a typical challenge in epidemiological studies, I also doubt how well these middle-aged participants can remember their body weight at 20 years of age. If authors are willing to report this, I recommend to express VO₂max as L/min.

All these findings include the same common feature: the higher is the body weight the higher is the odds ratio for having low CRF (ml/kg/min). The same exact problem is not present when studying the association between waist circumference (cm) and the risk of low CRF. However, this emphasizes the same fact that obese people tend to have low CRF.

Authors: Thank you for a relevant comment. We are aware of the problem regarding expressing CRF in relative terms in relation to BMI or previous weight (with both including kg). It was therefore we also included CRF in absolute terms (as L/min) in relation to BMI. Originally, we thought that it anyhow could add some info by presenting this information, but we have now concluded according to your comment to exclude both these correlations in the paper (both in main document and appendix).

Other comments:

Constant stress in men, and being a regular smoker in women, associated with lower risk of low VO2max. Could you please provide some discussion in relation to these findings? Are they just coincidence or could there be any other explanations for these contradictory findings?

Authors: Thank you for a relevant comment. We agree upon that these associations are somewhat contradictory to what could be expected. Previous experience has shown that regular smokers may have lower submaximal heart rate in relation to one work rate, hence resulting in higher estimated VO2max when performing submaximal tests relying on one heart rate response (such as the Åstrand-Rhyming cycle test). However, if the same phenomena is present for the test used (the Ekblom Bak test) has not yet been studied, and we are not able to speculate on this. The same goes for the association with high stress in men; whether this is due to a higher heart rate response on the lower, standardized work rate (resulting in a smaller difference in heart rate between the work rates, and a higher predicted VO2max) or unmeasured confounding related to higher socioeconomics in participants with higher/longer duration of perceived stress, are just two speculative possibilities. However, we choose to not include them in the manuscript due to the highly speculative nature.

Please provide explanations for the symbols a,b and c in the footnotes of the Table 3.

Authors: We have now added relevant footnotes for Table 3.

VERSION 2 – REVIEW

REVIEWER	Laukkanen, Jari A. Univ Eastern Finland, Institute of Clinical Medicine
REVIEW RETURNED	07-Nov-2022
GENERAL COMMENTS	The manuscript is improved, no further comments.
REVIEWER	Pentikäinen, Heikki Kuopio Research Institute of Exercise Medicine
REVIEW RETURNED	15-Nov-2022
GENERAL COMMENTS	I am satisfied with how authors have addressed to my worries concerning results presented in Table 2. Authors also provided nice discussion about the surprising association of stress in men and regular smoking in women with lower risk of low VO2max. I'm fine with the authors decision not to include the discussion to the manuscript due to the speculative nature of it. I don't have additional comments.